# In Vitro Modeling of Idiopathic Pulmonary Fibrosis: Lung-on-a-Chip Systems and Other 3D Cultures

**DOI:** 10.3390/ijms252111751

**Published:** 2024-11-01

**Authors:** Christopher Corona, Kun Man, Chad A. Newton, Kytai T. Nguyen, Yong Yang

**Affiliations:** 1Anne Burnett Marion School of Medicine, Texas Christian University, Fort Worth, TX 76129, USA; c.corona@tcu.edu; 2Department of Biomedical Engineering, University of North Texas, Denton, TX 76207, USA; kun.man@emory.edu; 3Department of Internal Medicine, University of Texas Southwestern Medical Center, Dallas, TX 75390, USA; chad.newton@utsouthwestern.edu; 4Department of Bioengineering, University of Texas at Arlington, Arlington, TX 76010, USA; knguyen@uta.edu

**Keywords:** idiopathic pulmonary fibrosis, lung-on-a-chip, in vitro models, disease models

## Abstract

Idiopathic pulmonary fibrosis (IPF) is a lethal disorder characterized by relentless progression of lung fibrosis that causes respiratory failure and early death. Currently, no curative treatments are available, and existing therapies include a limited selection of antifibrotic agents that only slow disease progression. The development of novel therapeutics has been hindered by a limited understanding of the disease’s etiology and pathogenesis. A significant challenge in developing new treatments and understanding IPF is the lack of in vitro models that accurately replicate crucial microenvironments. In response, three-dimensional (3D) in vitro models have emerged as powerful tools for replicating organ-level microenvironments seen in vivo. This review summarizes the state of the art in advanced 3D lung models that mimic many physiological and pathological processes observed in IPF. We begin with a brief overview of conventional models, such as 2D cell cultures and animal models, and then explore more advanced 3D models, focusing on lung-on-a-chip systems. We discuss the current challenges and future research opportunities in this field, aiming to advance the understanding of the disease and the development of novel devices to assess the effectiveness of new IPF treatments.

## 1. Introduction

Idiopathic pulmonary fibrosis (IPF) is a progressive and ultimately fatal interstitial lung disease characterized by the accumulation of fibrosis within the interstitial space of the lung parenchyma. IPF is associated with an extremely poor prognosis, with a median survival of only 3–5 years after diagnosis [1,2]. The exact mechanism underlying IPF is not fully understood, yet recent advancements in cellular biology and genomics have moved our understanding of the disease forward. It is theorized that IPF results from a complex interplay between genetic and environmental factors, leading to the activation of various signaling pathways and cellular processes that contribute to fibrosis development and propagation. At the tissue level, the alveoli face recurrent micro-injury by environmental stressors such as cigarette smoke, air pollution, silica dust, or pathogenic infection [3,4,5,6]. Some epigenetic factors, including altered DNA methylation and histone modifications, impact RNA-expression and promote fibrosis. Repeated injury induces progressive defective remodeling marked by excessive extracellular matrix (ECM) protein deposition within the interstitium. These cellular changes cause irreversible scarring and stiffening of the alveoli, leading to impaired gas exchange, decreased lung compliance, and eventual respiratory failure [7,8].

Lung transplantation is the only curative therapy for IPF but is only available for a small subset of patients. In 2014, the Federal Drug Administration (FDA) approved the therapeutics pirfenidone and nintedanib, which have been shown to slow the decline in lung function but do not cure the disease [9,10]. Prior to the approval of these antifibrotic drugs, clinicians often empirically used immunosuppressant agents such as corticosteroids and azathioprine. However, the PANTHER-IPF clinical trial demonstrated that these agents increased the risk of death and hospitalization [11]. The lack of highly effective pharmacological agents to treat IPF is largely attributed it’s complex and poorly understood pathophysiology Additionally, the stall of recent therapeutic advancements is due to a lack of clinically predictive models [12,13]. Animal models have traditionally been used to predict the in vivo pharmacodynamics of novel therapeutic agents. Specific to IPF, the bleomycin murine model attempts to serve as a bridge to human trials but is hindered by its low positive-predictive value and poor translation of human physiology [14,15]. While animal models can provide important insights into disease pathogenesis, caution is necessary when extrapolating animal study results to humans. Likewise, traditional two-dimensional (2D) monolayer cell cultures lack the necessary complexity for translational studies [16,17]. Therefore, it is clear that there is a need for in vitro models that mimic disease states at the organ, tissue, and cellular level. Thereby reliably predicting the in vivo pharmacology of novel therapeutics.

Three-dimensional (3D) cell culture models have emerged as a promising tool for elucidating pathophysiology in pulmonary disease. These models provide a more physiologically relevant platform to study complex microenvironmental interactions. Several categories of 3D cell culture models have been developed and deployed to study IPF, including hydrogels, precision lung slices, organoids, and lung-on-a-chip (lung chip) systems, each with distinct advantages and limitations (Figure 1). 3D models have enabled researchers to evaluate the efficacy and toxicity of potential drug candidates, aiding in the drug development process ofmany complex diseases [18]. Within the extensive landscape of 3D models, lung-on-a-chip models have emerged as a noteworthy system with significant potential. Lung chips can emulate in vivo lung microenvironments with high precision while concurrently assessing the impact of external mechanical forces. This review will provide an overview of the contemporary 3D models specifically designed for investigating pulmonary fibrosis. Subsequently, our focus will shift to the current advances in lung chips, underscoring their strengths and limitations. We will also highlight thier potential contributions to the study of IPF pathophysiology and their utility in the realm of therapeutic development.

## 2. 3D Cell Cultures: Modeling Pulmonary Fibrosis

In recent years, 3D in vitro cultures have gained popularity as an alternative to traditional 2D monolayer cultures for studying the complex pathogenesis of IPF and various other diseases. Although 2D cell cultures have been widely used in drug discovery for decades, they have several drawbacks when compared to 3D cell cultures. One major limitation of 2D cultures is that they fail to mimic tissue architecture and do not accurately represent the complex structure and organization of living tissues. Conversely, 3D cultures can replicate native tissue architecture, cell–cell communication, and cell–matrix interactions. Through the utilization of biomaterials and bioengineering techniques, 3D cell cultures provide customizable physiologically relevant models [17,19,20]. Additionally, the reduced cell–cell and cell–matrix interactions in 2D cultures lead to the loss of native differentiated states and functions over time, whereas 3D cultures can maintain signaling pathways and gene expression patterns [21]. Ultimately, 2D cell cultures offer overly simplistic cellular representation when a more robust model is needed. In contrast, 3D cell cultures, like hydrogels, organoids, and precision cut lung slices model complex cell interactions and tissue responses occurring in specific diseased states [22,23]. The highlighted 3D cell culture systems are summarized in Table 1 and discussed as follows.

### 2.1. 3D Hydrogels

Hydrogel-based in vitro cell cultures utilize 3D networks of hydrophilic macromolecules that are highly tunable. These macromolecular networks can be tailored to resemble the ECM found in specific tissues, including the lung [24]. Particularly in the context of pulmonary fibrosis, hydrogels can be engineered to mirror the biochemical composition and biomechanical properties of fibrotic lung tissue [25,26]. A significant advantage of using hydrogels in lung models is their ability to offer an environment that allows for studying cell–ECM interactions in both normal and diseased conditions, unlike traditional 2D cell cultures. For example, Asano et al. investigated the role of substrate stiffness in the activation and migration of lung fibroblasts, both critical factors in the pathophysiology of pulmonary fibrosis [27]. They found that increased substrate stiffness led to changes in fibroblast morphology, upregulated expression of α-smooth muscle actin (α-SMA), and enhanced cell migration. These results suggest the potential for substrate stiffness to promote fibrotic states. Likewise, additional studies have found that manipulating hydrogel stiffness can induce fibrotic states, revealing consequential impacts on gene expression and aberrant cellular responses [28,29,30,31].

Nevertheless, hydrogel-based 3D cultures are not without limitations. Despite their many benefits, they fail to incorporate external forces such as mechanical stretching, which restricts the study of cellular behavior under dynamic conditions. Hydrogels also display poor stability, resulting in degradative changes in gel properties over time that could alter cell behavior and tissue development [32,33]. Another major challenge is maintaining proper biological communication, particularly cell–cell interactions. The physical constraints and material stiffness of hydrogels often restrict these interactions leading to impaired signaling and reduced physiological relevance [34]. Improving these systems would require adjustments and alterations to the hydrogel matrix which would allow for appropriate system wide communication. Utilizing dynamic hydrogels that are tunable and reactive to stimuli can enhance cellular interactions. Recent advancements on this front have produced hydrogels that react to changes in pH, temperature, and lights therby producing physical and chemical alterations within the system [35]. Additionally, co-culturing multiple cell types and incorporating microfluidic perfusion systems can promote more natural signaling and intercellular communication.

### 2.2. Precision Cut Lung Slices

Precision cut lung slices (PCLS) are ex vivo models generated by slicing thin sections of lung tissues, usually 250–300 microns in thickness. PCLS are typically maintained in culture for several days while retaining many aspects of their in vivo structure and functions. One significant advantage of PCLS is that they preserve the lung tissue’s complex architecture, including the 3D organization of the airways, alveoli, and surrounding ECM [36]. The retention of in vivo architecture makes PCLS an ideal model for studying disease processes dependent on the complex interactions between different cell types and the ECM. Another advantage of PCLS is that they are be obtained directly from humans or animals, making them useful for translational research [37]. PCLS have also been used to study lung physiology, pharmacology, fibrosis and inflammation [38,39,40,41]. For instance, Alsafadi and colleagues induced fibrotic changes on PCLS from patients without interstitial lung disease [39]. This model exhibited early fibrotic-like changes upon exposure to profibrotic growth factors and signaling molecules. They demonstrated increased fibrotic gene expression within 24 h, significant protein level elevation at 48 h, increased ECM protein deposition, and alveolar epithelium reprogramming. This novel approach to modeling acute changes within native tissue is invaluable for elucidating the pathophysiology of IPF.

Regrettably, PCLS are very low throughput models and are relatively difficult to maintain viable in long-term culture [42]. These models require specialized equipment for generation and maintenance. In addition, there is high variability in slice quality, leading to inconsistent results among samples [36,43]. These limitations make them poor candidates for modeling disease progression in a longitudinal manner. Improving the viability and consistency of PCLS models requires advancements in automated slicing techniques and more precise control over culture conditions. Additionally, integrating perfusion systems or dynamic culture environments may extend the longevity of PCLS, allowing for better modeling of disease progression over time.

**Table 1 ijms-25-11751-t001:** 3D in vitro models, excluding lung chips, used to study pulmonary fibrosis.

3D Model	Cellular Composition	Applicability/Main Finding	Reference
Hydrogels	Human fibroblast (CCD-19lu) and primary fibroblast	FAK/Akt signaling promoting increased collagen deposition.	[25]
	Human lung fibroblast	Increased fibroblast activation and migration through matrix stiffening.	[27]
	Murine bleomycin treated lung fibroblast	PGE2 modulation of COX-2 suppression, fibroblast activation, and matrix stiffening.	[28]
	IPF and healthy human lung fibroblast	Matrix stiffening effects on proliferation, contraction, and resistance to PGE2.	[29]
	Primary human lung fibroblast	Pro-fibrotic stimuli hinder fibroblast apoptosis, altering Fas expression.	[30]
Precision Cut Lung Slices	Murine bleomycin lung slices	Protein biomarker utilization in drug screening.	[38]
	Healthy and IPF human lung slices	Modeling of early fibrosis.	[39]
	Human IPF lung tissue	Predictive markers of therapeutic response.	[40]
	Human/Murine bleomycin treated lung tissue	Differing response to Pirfenidone or Nintedanib by murine and human cultures.	[41]
Lung Organoids	Human pluripotent stem cells (hPSCs)	Modeling pulmonary fibrosis; antifibrotic assessment of potential therapeutic (MGF-E8)	[44]
	Murine mesenchymal and club cells	Mesenchymal support of bronchial organoid formation.	[45]
	Murine mesenchymal cells, macrophages, and bronchoalveolar stem cells	Branched bronchoalveolar organoid formation and modeling lung development.	[46]
	hPSC derived alveolar epithelial cells and primary human lung fibroblasts	Modeling pulmonary fibrosis: ALK5 and integrin aVb6 as therapeutic targets.	[47]
	Human alveolar basal cells	Bleomycin inducing honeycomb cyst formation	[48]

### 2.3. Lung Organoids

Organoids are 3D cell cultures derived from stem cells, supported by an embedded ECM, and can contain multiple cell types. The spatial arrangement that matches the in vivo structure can occur through self-organization or by spatially restricting the lineage commitments of stem cells. These systems have proven suitable for the investigation of mesenchymal–epithelial crosstalk and therapeutic testing [44,45,46]. Recent models have successfully recapitulated some key features observed in pulmonary fibrosis and identified potential therapeutic targets. For example, Suezawa et al. constructed an alveolar organoid derived from human pluripotent stem cells (hPSCs), alveolar epithelial cells, and lung fibroblast [47]. The organoid was subsequently exposed to bleomycin, a therapeutic chemical agent that induces fibrosis. Post exposure analysis found that the model replicated features specific to the initiation of pulmonary fibrosis. In particular, the organoid exhibited fibroblast activation, cellular senescence, abnormal cell differentiation, and ECM accumulation. The model identified potential therapeutic protein targets, namley, ALK5 and αVβ6 integrins. Inhibition and blocking of these proteins demonstrated notable improvements in fibrogenic changes. More recent organoid models have shown the ability to replicate pathological structural changes within the lung, such as honeycomb cyst formation, a distinct histopathological feature of IPF [48].

Early iterations of lung organoid models were limited by their inability to integrate vasculature structures, immune cell interactions, and physiologic mechanical stressors [49]. The complexity and heterogeneity of organoids also posed challenges for standardization as compared to other 3D models. However, recent advancements, such as AggreWell technology, have improved organoid reproducibility and scalability. AggreWell platforms use microwell arrays to guide the aggregation of stem cells or other progenitor cells, ensuring that organoids are generated with uniform size and structure [50,51]. Complementary approaches, such as microarray 3D bioprinting, enable precise cell placement and dynamic culture conditions, further enhancing organoid utility [52]. These innovations simplify the differentiation process, making organoid generation more consistent and suitable for high-throughput screening (HTS). Notably, newer systems integrate dynamic flow conditions and microfluidic devices to further standardize organoid development [53,54].

## 3. Lung-on-a-Chip Systems

Several advances in bioengineering and microfluidics have coalesced to enable the creation of highly complex lung-on-a-chip systems. These customizable 3D cell culture systems use microfluidic technology to create intricate microchannel networks that mimic organ- and tissue-level physiology. The first step in chip design involves identifying the microenvironment of interest, this guides the selection of cell types and informs the structural layout of the chip. Lung chips are uniquely capable of replicating key structural and functional features of lung parenchyma, including cell–cell communication, air–liquid interfaces, vascular perfusion, and multi-cell type co-culture. For example, some chips have been designed to specifically replicate the alveolar–capillary interface of the lungs, replicating how gas exchange occurs in the body [55,56,57].

Traditionally, soft lithography techniques are used to fabricate the intricate networks of channelswithin lung chips. 3D printing technology has also emerged as an alternative option, facilitating rapid prototyping and an iterative design process [58,59]. The specific design and configuration of channels within lung chips can vary depending on the desired applications. For instance, some lung chips incorporate microfluidic valves or pumps to mimic breathing motions or fluid flow patterns within the lung [60]. Others employ more complex channel geometries to investigate the effects of mechanical stress on lung tissues or to replicate the pathophysiology of lung diseases [55,61]. Materials used in lung chips are chosen for their biocompatibility, gas permeability, and mechanical properties. Once the design is finalized, lung chips are typically constructed from biocompatible materials, such as polydimethylsiloxane (PDMS), chosen for its gas permeability, elasticity, and optical transparency. These properties allow for both real-time observation of cellular responses and the replication of lung tissue stretch. Although PDMS is widely used, recent developments have introduced alternatives like collagen–elastin composites. These biomimetic membranes better emulate the extracellular matrix (ECM) stiffness and mechanical properties observed in healthy or diseased lung tissue, providing improved physiological relevance compared to synthetic polymers [62]. Furthermore, our team has previously published a detailed review on the construction of lung-chip systems, including their engineering principles, materials, and design considerations for those readers interested in more technical details [63] (Figure 2).

A critical aspect of lung chip models is maintaining the integrity of the alveolar–capillary barrier, as its disruption is a hallmark of many lung diseases. To assess barrier function, researchers often perform permeability assays using markers such as FITC-dextran to quantify the diffusion of molecules across the epithelial and endothelial layers. Immunofluorescence staining for tight junction proteins (e.g., occludins and ZO-1) is also used to visualize the formation and integrity of intercellular junctions. Validating lung chips involves benchmarking against physiological and pathological lung responses observed in vivo. Functional assays, such as surfactant secretion analysis, respiratory mechanics under cyclic stretch, and cytokine release following exposure to pathogens or environmental toxins, are employed to evaluate the accuracy of these models. Additionally, comparison to clinical data from human lung tissues or biopsies enhances the translational relevance of these models for drug testing and disease modeling.

Lung chips enable researchers to investigate lung pathologywithin a physiologically representative and dynamic environment. Moreover, the ability to co-culture multiple cell types within the same device allows for recreating intricate cellular interactions that are impossible in conventional cell culture models. In the subsequent sections, we will delve into the latest advancements in lung chip technology and explore its application as an in vitro model of pulmonary fibrosis. The highlighted lung chip models are summarized in Table 2.

**Table 2 ijms-25-11751-t002:** Recent lung chip systems for modeling pathophysiologic states, lung injury, fibrotic response, and therapeutic testing.

Modeling Approach	Cellular Composition	Key Outcome	References
Normal Physiologic State	Human alveolar epithelial and pulmonary endothelial cells	Inflammatory, immune, and stress reactions to pollutants.	[64]
	Airway epithelial cells, lung fibroblast, and endothelial cells	Multi-layered, co-cultured replicating cellular composition of lung.	[65]
	Type I and II alveolar epithelial cells, endothelial cells	Medium-throughput physiologic three-dimensional stretching system	[66]
	Alveolar epithelial and endothelial cells	collagen-elastin membrane replicating geometric and biophysical characteristics of ECM	[62]
	Alveolar epithelial cells and fibroblasts	Enhanced nano spun pseudo-interstitium, improving epithelial barrier function and longevity.	[67]
Alveolar Injury	Human type II A549	Exposure to gastric contents induced cellular injury.	[68]
	Human pulmonary alveolar epithelial and umbilical vascular endothelial cells	Nanoparticles cause dose-dependent toxicity to lung cells	[69]
	Human alveolar epithelial cells, pulmonary vascular endothelial cells, and human acute leukemia monocytic cell lines (THP-1)	Air pollutants disrupt alveolar-capillary interface, induce inflammation, and trigger immune cell recruitment.	[70]
	Immortalized human alveolar epithelial cell lines (iAECs), primary human lung microvascular endothelial cells, and peripheral blood mononuclear cells (PBMCs).	Bacterial endotoxin exposure causing alveolar barrier disruption and inflammation.	[71]
	Human primary alveolar epithelial cells and human lung microvascular endothelial cells (HMVEC-L)	Radiation induced lung injury. Therapeutic response to lovastatin and prednisolone.	[72]
Induction of Fibrosis	Pulmonary Fibroblast	Simulate fibrotic events, predicts antifibrotic effects of pirfenidone and nintedanib	[73]
	Primary human lung microvascular endothelial cells iAECs, and PBMCs	TGFβ1 induced epithelial to mesenchymal transition.	[71]
	Human-induced pluripotent stem cell-derived endothelial cells (hiPSC-ECs) and primary human lung fibroblast.	Stromal-endothelial interactions modulate changes in vessel density, expression levels, and tissue stiffness.	[74]
Therapeutic Testing and Drug Delivery	Primary human lung fibroblasts	Antifibrotic drug efficacy.	[75]
	Primary cell-derived immortalized alveolar epithelial cells (AXiAECs), macrophages (THP-1), and endothelial (HLMVEC) cells	Aerosol delivery system and inhaled steroid efficacy.	[76]

**Figure 2 ijms-25-11751-f002:**
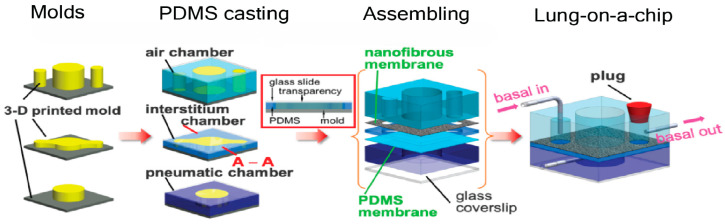
Representative overview of lung-on-a-chip fabrication. Reprinted/adapted with permission from [67]. Copyright 2024 American Chemical Society.

### 3.1. Physiologic Memetic Models

#### 3.1.1. Essential Cell Types and Physiologic Stretch

Lung chip models that accurately replicate normal physiologic states are crucial to advancing our understanding of complex organ-level responses. Before delving into models of pathologic states, it is imperative to establish these physiologically relevant models. By effectively mimicking lung microenvironments and cellular interactions, such models provide a robust framework for studying pulmonary fibrosis. In 2010, Huh et al. reported for the first time a microfluidic device that recapitulates the functional alveolar–capillary interface of the human lung [64]. This device integrated a novel vacuum-controlled cyclical mechanical stretching system to mimic physiological breathing. More importantly, this device reconstituted multiple complex organ-level responses. For example, when exposed to pathogenic bacteria, the model induced an inflammatory cytokine response from the epithelial layer, resulting in endothelial activation and the transmigration of neutrophils across the alveolar–capillary barrier. Additionally, the model demonstrated the adverse effect of mechanical stress by physiologic breathing in the presence of air pollutants. They measured a substantial increase in reactive oxygen species (ROS) and pro-inflammatory adhesion molecules upon eposure to toxic nanoparticles. This work would serve as the framework for future models working towards recapitulating more accurate microenvironments. Another important aspect in developing effective lung chip models is incorporating multiple cell types within a single device thereby mimicking the cellular composition of native lung tissues. To this end, Sellgren et al. developed a model incorporating airway epithelial cells cultured at an air–liquid interface with lung fibroblast cells and polarized microvascular endothelial cells [65]. These three distinct layers were vertically stacked and separated by a nanoporous membrane, allowing for co-culture. Permeability testing within the cellular layers successfully indicated the formation of cell junctions and the presence of a cellular barrier. Immunostaining and confocal microscopy also confirmed physiologically accurate phenotypes of all three cell layers. In addition to incorporating multiple cell types within a system, the integration of cyclic stretching in lung chip models further enhanced the models physiological relevance. Stukci et al. developed a medium-throughput device that incorporates six individual lung chips, each containing fully differentiated primary Type I and II alveolar epithelial cells and endothelial cells [66]. Their chip includes a cyclic vacuum cavity that mimics the directionality of diaphragm contraction by allowing for 3D stretching as opposed to the 2D stretching seen in previous models. This device demonstrated a unique ability to replicate key aspects of the alveolar microenvironment in a “breathing” system while maintaining the formation of an air–liquid interface, as measured by surfactant production and barrier permeability analysis.

#### 3.1.2. Modeling the Interstitial Space

As much of the pathophysiology of pulmonary fibrosis unfolds within the interstitial space, particular emphasis has been placed on the micro engineered layers representing the ECM within lung chip models. These bioengineered layers serve as crucial supportive structures and ECM surrogates. In their innovative approach, Zamprogno et al. replaced the conventional PDMS membrane with a novel collagen–elastin (CE) membrane of similar composition to that seen in vivo [62]. This CE membrane is supported by a gold hexagonal mesh, which is initially poured as a liquid and then kept in place on the mesh through surface tension. With subsequent heating and gelling the CE membrane becomes firmly anchored to the gold mesh, allowing for seeding of epithelial and endothelial cells on both the basal and apical sides. Notably, the novel CE membrane demonstrated remarkable similarities to the ECM observed in vivo, effectively replicating the geometrical, biophysical, and transport properties associated with the alveoli. Man et al. developed a lung chip model with a specific focus on the composition of the pseudo-interstitium [67] (Figure 3a,b). This approach involved incorporating an electrospun nanofibrous membrane to co-culture the epithelial cells and fibroblasts encapsulated in a collagenous gel. This unique interstitium matrix effectively interfaced with the epithelial cells and allowed for a physiological media flow throughout the system. The biomimetic chip demonstrated substantial improvements in epithelial barrier function when compared to traditional transwell models. Moreover, utilizing a collagen I-fibrin blend within the interstitial matrix extended the chip longevity beyond eight weeks.

These representative devices showcase the capacity of current models to optimize and replicate the intricate elements of the cellular microenvironment. The framework established by modeling normal physiology on lung chip systems has the potential to serve as a control in future experimental studies examining cellular response to therapeutics, injury, and toxins. However, researchers face the challenge of balancing features like system complexity, longevity, and throughput in their work.

### 3.2. Modeling Alveolar Injury

Numerous studies have suggested that environmental factors such as smoking, occupational hazards, gastroesophageal reflux, and viral infections are principal contributors to such injuries [3,4,6]. Therefore, unopposed repeated alveolar injury plays a key role in the advancement and progression of IPF. However, identifying a single predominant causative agent remains elusive. Investigators utilizing lung chip systems actively pursue this objective by creating sophisticated models to delineate the mechanisms underlying lung injury.

#### 3.2.1. Endogenous Cellular Injury

For instance, Felder et al. used epidemiological data to guide their chip design and simulated micro-injuries in human type II A549 cell lines by exposing the cells to trypsin and gastric content [68]. They hypothesized that micro-aspirations of gastric contents, occurring in gastroesophageal reflux disease, may play a significant role in the progression of IPF. After subjecting the cells to this exposure, they observed compacted cell nuclei, which is indicative of DNA fragmentation due to apoptosis. This observation aligns with histologic findings in IPF patients. Their team concluded that HCl exposure was responsible for the observed cellular damage, thereby verifying the hypothesis that gastric content may contribute to IPF.

#### 3.2.2. Exogenous Pollutants Causing Cellular Injury

In a similar experiment aimed at assessing the harmful effects of nanoparticles on the alveoli, Zhang and colleagues devised a parallel three-channel lung chip [69]. This chip was comprised of human pulmonary alveolar epithelial cells and human umbilical vascular endothelial cells with the intent of representing the interface between the alveoli and capillaries. To mimic the ECM, a layer of Matrigel was placed between these two cell lines. Furthermore, the endothelial cells in the vessel channel were subjected to a fluidic flow, replicating the blood flow observed in vivo. The alveoli chip was then exposed to two types of nanoparticles on the apical side: TiO_2_ NPs (particles considered safe for human ingestion and used in food color additives, medicines, and architecture industries) and ZnO NPs (known for their toxicity to mammalian cells and utilized in the medical and dietary sectors). Both TiO_2_ and ZnO NPs demonstrated dose-dependent toxicity towards the epithelial and endothelial layersas evidenced by the generation of ROS and induction of apoptosis. This experimental system provided a reliable model for studying nanoparticle induced pulmonary injury, which is crucial for understanding the contribution of occupational and environmental exposures to the development of IPF. Similarly, Xu et al. modeled the induction of alveolar injury through exposure to fine particulate air pollutants (PM2.5) [70] (Figure 3c,d). Their device featured human alveolar epithelial cells and pulmonary vascular endothelial cells separated by a Matrigel layer mimicking the ECM. Each cell layer was flanked by a microfluidic channel; culture media and human acute leukemia monocytic cell lines (THP-1) flowed through the vessel side while pollutant treatments were introduced on the alveolar side. They demonstrated that exposure to pollutants disrupted the alveolar-capillary interface, induced ROS generation, increased the expression of interleukin (IL)-6 and tumor necrosis factor (TNF)-a. Additionally, they measured the notable attachment of THP-1 cells to the damaged endothelial layers, indicating immune cell recruitment.

#### 3.2.3. Infectious Agents

Recurrent bacterial infections are also believed to potentially contribute to the development of pulmonary fibrosis. To explore this connection, Sengupta et al. designed an alveolar chip exposed to bacterial lipopolysaccharide aimed to simulate various stages of lung injury associated with recurrent bacterial infections [71]. By co-culturing immortalized human alveolar epithelial cell lines (iAECs), primary human lung microvascular endothelial cells, and peripheral blood mononuclear cells (PBMCs), they accurately replicated the inflammatory responses to bacterial pathogens. The findings revealed a significant disruption of the alveolar barrier and increased levels of the proinflammatory cytokine IL-8. These results suggest that the model is suitable for studying the impact of recurrent bacterial damage in pulmonary fibrosis.

#### 3.2.4. Radiation Effects

Acute radiation exposure is known to cause radiation-induced lung injury (RILI). Dasgupta et al. created a lung chip system lined with human alveolar epithelial and pulmonary microvascular endothelial cells [72]. This system effectively demonstrated radiation-induced DNA damage, cellular hypertrophy, and increased inflammatory markers. It provides a human-relevant platform for studying RILI and the effectiveness of antifibrotic therapeutics, lovastatin, and prednisolone. Overall, current lung chip systems have shown the capability to accurately simulate suggested injury mechanisms and tissue responses. As lung chip models progress and become capable of sustaining cell cultures for longer durations, it will be essential to examine the impact of persistent injury and repair. Meanwhile, expanding on injury studies to include other hypothesized etiologies remains prudent.

#### 3.2.5. Inducing a Fibrotic State

Central to the pathogenesis of IPF is the initiation and progression of fibrosis, which results in scarring and stiffening of the alveoli. Understanding and modeling this increased proliferation of myofibroblast and subsequent dysfunctional ECM remodeling is therefore highly relevant. For example, Asmani et al. developed a novel class of fibrotic microtissue arrays [73]. Their device simulates key biological events occurring during pulmonary fibrosis, including increased tissue stiffness and decreased tissue compliance. Their interstitium-like microtissue combines a fibroblast-populated collagen matrix suspended on PDMS micropillars, resembling alveolar sacs in surface area and thickness. Treatment with TGFβ1 resulted in a biomimetic increase in fibrotic phenotype markers like α-SMA, pro-collagen, and fibronectin. Notably, the analysis demonstrated a significant decrease in compliance and an increase in tissue stiffness after TGFβ1 treatment, which are key features of the functional lung decline observed in vivo. Furthermore, their device reliably predicted the antifibrotic effects of pirfenidone and nintedanib. While this microtissue array device provides a relatively high-throughput platform for accurate fibrosis modeling and therapeutic testing, the addition of other representative tissue layers would enhance its in vivo predictability. As an adjunct to their lipopolysaccharide injury model, Sengupta et al. modeled TGFβ1-induced fibrosis within their device. They observed a decrease in epithelial-specific markers after TGFβ1 treatment and an increase in mesenchymal cell targets such as ACTA2 and COL1A1, suggesting an epithelial-to-mesenchymal transition (EMT) [71]. In another study conducted by Akinbote and colleagues, a perfusable vascular chip was designed to investigate the role of stromal–endothelial crosstalk following the induction of fibrosis with TGFβ1 [74]. Human-induced pluripotent stem cell-derived endothelial cells (hiPSC-ECs) were cultured in the presence and absence of primary human lung fibroblasts. They observed that the presence of lung fibroblasts markedly reduced the microvessel density, decreased matrix metalloproteinase (MMP) 1 and 9 expression, increased tissue stiffness, and upregulated α-SMA expression. This device highlights the key role of stromal–endothelial interactions in the fibrotic phenotype. Accurately reproducing the fibrotic response is vital for advancing models intended for therapeutic testing.

The existing lung chip models demonstrate their ability to simulate fibrotic states effectively at both the tissue and cellular levels. To enhance the applicability of these models for therapeutic testing, future studies should focus on incorporating representative cell types from all layers of the lung parenchyma, circulating immune cells, and vascular structures. This comprehensive approach will prove invaluable in advancing lung chip therapeutic research.

### 3.3. Therapeutic Testing and Drug Delivery

Current preclinical models for drug development have poor predictive power, leading to costly delays in development and approval of therapeutics. Advancements in pathophysiologic modeling for lung fibrosis have paved the way for a new frontier within in vitro therapeutic testing and drug delivery systems. Microtissue array systems have previously been used to screen FDA-approved anti-fibrosis drugs pirfenidone and nintedanib, demonstrating their efficacy in reducing fibrosis in vitro [73]. Recently, this same model was expanded upon and used to test the efficacy of two novel anti-fibrosis drugs under clinical trials, KD025 and BMS-986020. The system found that the novel therapeutic candidates were able to inhibit myofibroblast activation, as well as cell-mediated ECM contraction. Furthermore, the novel drugs had comparable efficacy to pirfenidone and nintedanib [75]. Sengupta et al. designed a new generation platform which integrates a lung chip system with an exposure chamber that allows for the aerosol delivery of fibrotic inducing agents and therapeutics, dubbed the Cloud α AX12 [76]. Their model induced fibrosis through exposure to polyhexamethylene guanidine (PHMG). Subsequent treatment of the fibrosis chip with inhaled corticosteroid fluticasone showed a reduction in the expression of inflammation and epithelial damage-associated genes, demonstrating its utility as a model for aerosol drug delivery. The lung chip system’s versatility as a model for therapeutic testing underscores its potential to serve as a robust preclinical tool. However, for it to reach this potential, limitations in standardization and integration with existing regulatory frameworks must be successfully addressed.

## 4. 3D Model Applications Beyond Pulmonary Fibrosis: Insights into Lung Injuries

While this review focuses on idiopathic pulmonary fibrosis, 3D in vitro models have also been valuable for studying other lung injuries. These models offer insights into the pathophysiology of conditions such as infections, asthma, and lung cancer. Alveoli-on-a-chip platforms simulate inflammation in pneumonia models, allowing the investigation of immune responses and endothelial injury, as demonstrated in studies on influenza and COVID-19 [77,78]. Influenza models, for instance, showed ciliary dysfunction and increased cytokine production, while COVID-19 models helped identify potential antiviral therapies by testing FDA-approved drugs in systems expressing human ACE2 and SARS-CoV-2 spike proteins. Small airway models of asthma replicate key disease features, including airway remodeling and inflammatory responsesOrganoids derived from asthmatic patients mimic increased mucus production and altered tight junctions when exposed to dust mites [79]. Lung cancer-on-a-chip models aid in evaluating tumor growth, drug response, and immune interactions, with studies demonstrating angiogenesis and tumor invasion [80]. Hydrogel systems have been utilized to demonstrate the decreased response and migration of Natural Killer-92 cells within systems containing metastatic A549 cell lines as compared to healthy controls [81]. Lung tumor-on-a-chip systems have demonstrated the ability to map and monitor the progression of cancer cell death as a result of chemotherapy, and how it compares to natural cell death [82]. Moreover, this model found that chemotherapy-induced cell death promoted the death of nearby cancer cells, something not seen in natural cell death pathways. 3D in vitro models not only offer innovative ways to study the pathophysiology of a wide range of lung diseases but also provide critical insights that can drive the development of more effective therapeutics.

## 5. Outlooks

IPF remains the most severe diffuse interstitial lung disease. Although the mechanism of IPF has been studied for many years, much remains unknown about the molecular drivers of this disease. Current 3D in vitro models, particularly lung chips, have demonstrated suitability in recapitulating IPF microenvironments. The lung chips evolved from initial designs that aimed to replicate the alveolar–capillary interface to sophisticated models that integrate multiple cell types and mimic various environmental and pathological conditions. They have tremendously advanced our understanding of lung disease mechanisms at the organ level. Innovative approaches, such as replicating ECM properties, simulating various injury mechanisms, and emulating fibrotic responses, have illuminated the intricacies of pulmonary fibrosis. These models provide insights into disease progression, the role of different cell types, the impact of mechanical stress, and the contribution of environmental factors. Lung chip platforms are laying a foundation for new hypotheses and potential therapeutic strategies.

However, lung chips still face several limitations that must be addressed to fully realize their potential. A major issue is the lack of standardization in design and fabrication, leading to variability in results across different studies [83]. Improved standardization, especially in chip design and operating procedures, could enhance reproducibility and enable more consistent research outcomes. Moreover, most lung chips, though highly complex, still oversimplify key aspects of lung physiology such as vascular networks and immune responses. Particularly, the involvement of multiple cell types and complex cytokine signaling are missing from most systems. Some lung chip systems have demonstrated immune responses through neutrophil and macrophage recruitment; still, this is only a small part of the robust immune response seen in vivo [64,84]. Incorporating more advanced vascular structures and multiple immune cell interactions could significantly enhance the physiological relevance of these models, improving their utility for drug testing and disease modeling.

Another challenge with lung chips lies in imaging limitations. As systems become more complex and three-dimensional, traditional imaging techniques used in 2D cultures may no longer suffice. Developing advanced imaging methods that can track morphological and histological changes within 3D environments would be essential. Additionally, materials like PDMS, while offering gas permeability and optical transparency, can interact with small molecules and drugs potentially skewing experimental results [85]. Future systems might benefit from alternatives biomaterials that provide better biomimicry without compromising experimental outcomes.

Looking toward the future, lung chip systems are poised to advance significantly in terms of system longevity and complexity. Enhancement in longevity will enable chronic disease modeling and allow for the long-term observation of disease progression and treatment. The extended chip longevity can be achieved by dynamic perfusion systems that optimize nutrient delivery and waste management. Providing this longitudinal view would enable researchers to map the disease trajectory, identifying key stages where structural and functional changes occur in the organ. This can lead to the development of predictive models for disease progression and enable clinicians to provide more accurate prognostic information to patients. Thus, this approach may help identify temporal therapeutic windows—specific points in disease progression where therapeutic intervention could be most effective. Understanding when these windows open and close can potentially lead to more precise treatments.

Furthermore, modifying lung chip models through the utilization of patient-derived cells has the potential to advance personalized medicine. This personalized approach recognizes the biological uniqueness of each patient and acknowledges the varied natural history and therapeutic response between individuals [86,87,88]. Establishing personalized chips could pave the way for more precise and effective individualized treatments, reducing trial-and-error prescribing and increasing the likelihood of successful outcomes. Additionally, lung chip technology could leverage advances in genome-wide studies to further refine personalized systems [89,90].

In summary, lung-on-a-chip technology offers a powerful platform for studying pulmonary fibrosis and other lung diseases. Further advances in these systems will not only improve our understanding of lung pathophysiology but also accelerate the development and evaluation of novel therapeutic interventions, potentially transforming patient outcomes. The future is indeed promising for lung-on-a-chip technology, and its optimal realization could revolutionize our approaches to lung disease research and treatment.

## Figures and Tables

**Figure 1 ijms-25-11751-f001:**
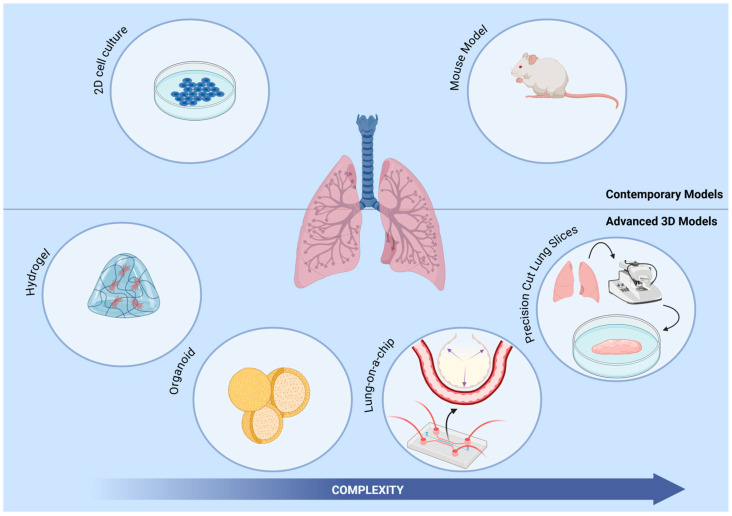
Various 3D human systems assist in IPF research. This includes hydrogel cultures of lung fibroblasts for cell interaction studies and therapeutic testing. Patient-derived precision-cut lung slices for modeling various stages of fibrosis and drug screening. Lung organoids from stem cells for modeling IPF pathophysiology and therapeutic testing. Lung-on-a-chip systems incorporate multiple cell types for modeling IPF, recapitulate epithelial and vascular functions, and apply dynamic mechanical strain.

**Figure 3 ijms-25-11751-f003:**
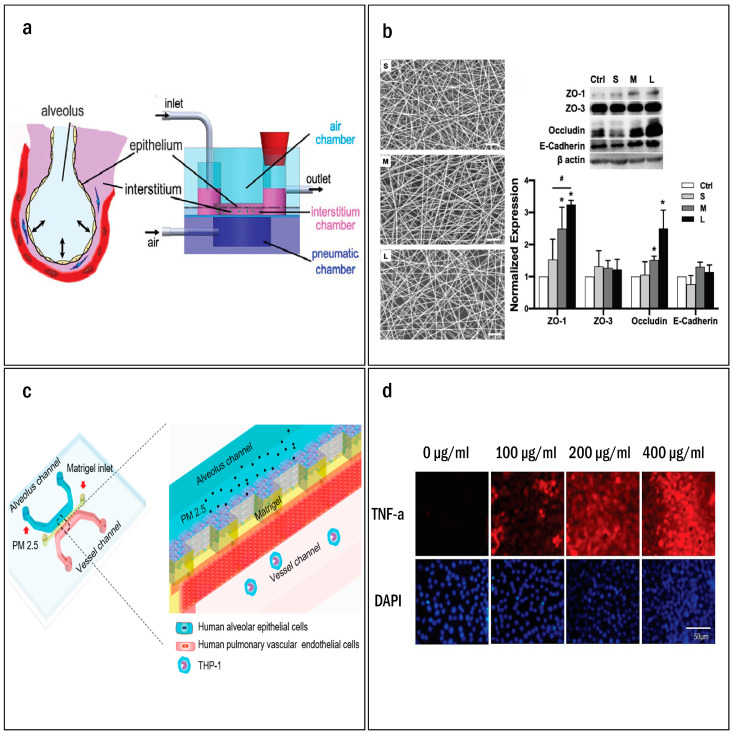
(**a**) Schematic illustration of human lung alveolus electrospun nanofibrous lung-on-a-chip system. (**b**) Scanning electron micrographs of nanofibrous membranes of small (S), medium (M), and large (L) pore sizes. Scale bars (white): 20 μm. Western blotting of ZO-1, ZO-3, occludin, and E-cadherin (epithelial barrier makers) of A549 cells cultured on the flat control and nanofibrous membranes with S, M, and L pore sizes (*n* = 3). The data were normalized to the mean value of the flat control. *: *p* < 0.05 compared to the flat control. #: *p* < 0.05 between groups. (**c**) Design and structure overview of the lung-on-a-chip system for PM2.5 exposure. This device is composed of three parallel channels: mimicking the ECM membrane between the epithelium and endothelium in vivo. (**d**) Representative immunofluorescence of TNF-α expressing in HPAEpiCs. After 3 days of exposure to PM2.5, HPAEpiCs were fixed and immunostained with TNF-α antibody. Reprinted/adapted with permission from Refs. [67,70]. Copyright 2024 American Chemical Society.

## Data Availability

Not applicable.

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
