# Peer review of "In Vitro Modeling of Idiopathic Pulmonary Fibrosis: Lung-on-a-Chip Systems and Other 3D Cultures"

_ijms, 2024, doi:10.3390/ijms252111751_

Round 1
Reviewer 1 Report
Comments and Suggestions for Authors
1. 3.2.1. is it A549 or A459 cell line?
2. Can you write short summary on how 3D models are used in other lung injuries? May be a table?
3. Please cross verify all the references and add where needed
Author Response
Response : Thank you for pointing out these typos! They are A549 cell line. The manuscript now reflects the correct cell line. |
Comments 2: Can you write short summary on how 3D models are used in other lung injuries? May be a table? |
Response : Section 4 “3D Model Applications Beyond Pulmonary Fibrosis: Insights into Lung Injuries” has been added. This section highlights how 3D models have been utilized for other lung injuries, including lung infections (influenza and SARs-CoV-2), asthma, and lung cancer. It provides an overview with cited examples for each pathology. Comments 3: Please cross verify all the references and add where needed Response : Certainly, we have reviewed all references and added some with the addition of the new summary paragraph.
|
Reviewer 2 Report
Comments and Suggestions for Authors
The article provides an overview of current in vitro 3D modeling of Idiopathic Pulmonary Fibrosis and analyzes the limitations and progress of each approach. The content is logically organized and comprehensive. The reviewer suggests minor revisions before publication, with the following main concerns:
1. When discussing the limitations of hydrogels, it is important to note that one significant challenge is maintaining biological communication, such as cell-cell interactions, in 3D cultures within hydrogels. This is an area that could be considered for improving the method.
2. Recent studies have used the AggreWell plate for constructing 3D models. Should this be included in the discussion?
3. The 3D methods mentioned each have their limitations. How can these models be improved or enhanced in future research?
Comments on the Quality of English LanguageGood
Author Response
Comments 1: When discussing the limitations of hydrogels, it is important to note that one significant challenge is maintaining biological communication, such as cell-cell interactions, in 3D cultures within hydrogels. This is an area that could be considered for improving the method. |
Response : Thank you for the suggestion! We have expanded on the limitations for hydrogels to maintain biological communications within the system in section 2.3 “3D Hydrogels”. We have also offered additional suggestions to improve these systems through manipulating physical constraints (material stiffness), utilization of hydrogels that are dynamic/responsive to stimuli and by incorporating microfluidic channels. [Paragraph 2, Line 124]
|
Comments 2: Recent studies have used the AggreWell plate for constructing 3D models. Should this be included in the discussion? |
Response : We appreciate the suggestion! Aggrewell technology was included in our discussion regarding lung organoids in Section 2.3 “Lung Organoid”. We highlighted how this technology might help with improving reproducibility and scalability concerns. We paralleled their advantages with microwell 3D bioprinting technique which also strives to standardize organoid systems. [Paragraph 2, Line 182]
Comments 3: The 3D methods mentioned each have their limitations. How can these models be improved or enhanced in future research? Response : We have added sentences discussing automated slicing techniques and dynamic culture environments to improve consistency and longevity of PCLS models for better disease modeling. [Section 2.2, paragraph 2, line 158].
The response in comments 1 and 2 address improvements for Hydrogels and Organoids.
We have expanded [Section 5, “Outlooks”] to address the current limitations of lung-on-a-chip models, including the lack of standardization, oversimplification of vascular and immune responses, and challenges with imaging techniques. We emphasize that improved standardization in design and operating procedures will enhance reproducibility, and we propose that future advancements should focus on incorporating more complex vascular structures and immune cell interactions to improve physiological relevance. [Paragraphs 2 & 3, Line 496]. Additionally, we discuss the need for advanced imaging methods capable of tracking changes in 3D environments and suggest alternatives to PDMS, such as collagen-elastin composites, to minimize experimental interference future developments, including dynamic perfusion systems for better nutrient delivery, are highlighted as key to extending system longevity and enabling long-term disease modeling Paragraph 4, Line 508]. We also propose that integrating patient-derived cells will enhance personalized medicine by allowing for more tailored treatments [Paragraph 5, Line 528].
|
Reviewer 3 Report
Comments and Suggestions for Authors
In this paper, the authors reviewed the In Vitro Modeling of Idiopathic Pulmonary Fibrosis: Lung-on-a-Chip Systems and Other 3D Cultures. Overall, it is well-written, but the following comments should be addressed before further processing:
1- More details should be provided on the engineering aspects of lung-on-a-chip models, including chip design, components, materials used, barrier integrity assessments, and the validation of these models.
2- It is highly recommended to include 2 or 3 multi-pane figures from the discussed examples, specifically for the 3D culture systems and lung-on-a-chip models from previous studies. The figures should also include proper permissions from the publishers in the captions.
3- The ( outlooks ) section on challenges and future directions in lung-on-a-chip model development should be improved. The authors should discuss the limitations of current models and elaborate on potential future advancements to overcome these challenges.
Comments on the Quality of English LanguageThe English can be improved
Author Response
Comments 1: More details should be provided on the engineering aspects of lung-on-a-chip models, including chip design, components, materials used, barrier integrity assessments, and the validation of these models |
Response : Thanks for the valuable suggestion! We made substantial revisions accordingly in Section 3. We now described how the initial design process involves identifying the target microenvironment to guide cell selection and structural layout [Paragraph 1, Line 196]. We also elaborated on fabrication techniques, such as soft lithography and 3D printing, to create microchannel networks that mimic lung physiology [Paragraph 2, Line 207]. Regarding materials, we provide additional information on the commonly used PDMS and the introduction of collagen-elastin composites to better emulate ECM stiffness and mechanical properties [Paragraph 2, Line 216]. For barrier integrity, we explain methods like FITC-dextran permeability assays and immunofluorescence staining for tight junction proteins [Paragraph 3, Line 236]. To validate the models, we discuss functional assays, including surfactant secretion and respiratory mechanics, and highlight how comparison to clinical data enhances their relevance [Paragraph 3, Line 241]. For readers seeking more in-depth technical details, we direct them to our previously published review article, which offers a comprehensive exploration of lung-chip system construction, materials, and design considerations [Paragraph 2, Line 224]. |
Comments 2: It is highly recommended to include 2 or 3 multi-pane figures from the discussed examples, specifically for the 3D culture systems and lung-on-a-chip models from previous studies. The figures should also include proper permissions from the publishers in the captions. |
Response : Figure 2 was added to the manuscript. Figure 2(a) is a representative image for the fabrication process of lung-on-a-chip system. Figure 3(a) illustrates how the fabrication process can be applied to study a particular lung injury (i.e., pollutant exposure). The rights to use these figures was obtained from their publisher. The caption reflects the preferred method of citation based on the publishers instruction.
Comments 3: The ( outlooks ) section on challenges and future directions in lung-on-a-chip model development should be improved. The authors should discuss the limitations of current models and elaborate on potential future advancements to overcome these challenges. Response : We have expanded Section 5, “Outlooks” to address the current limitations of lung-on-a-chip models, including the lack of standardization, oversimplification of vascular and immune responses, and challenges with imaging techniques. We emphasize that improved standardization in design and operating procedures will enhance reproducibility, and we propose that future advancements should focus on incorporating more complex vascular structures and immune cell interactions to improve physiological relevance. [Paragraphs 2 & 3, Line 495]. Additionally, we discuss the need for advanced imaging methods capable of tracking changes in 3D environments and suggest alternatives to PDMS, such as collagen-elastin composites, to minimize experimental interference. Adding dynamic perfusion systems for better nutrient delivery, are highlighted as keys to extending system longevity and enabling long-term disease modeling Paragraph 4, Line 508]. We also propose that integrating patient-derived cells will enhance personalized medicine by allowing more tailored treatments [Paragraph 5, Line 528].
|
Round 2
Reviewer 3 Report
Comments and Suggestions for Authors
The authors have made efforts to address my comments; however:
-
The figures still lack the visual appeal needed to engage readers. Given the many interesting studies in this field, such as those showcasing the channel structures in breathing lung-on-a-chip models, it would be beneficial to include similar visual elements in Figures 1 and 2.
-
Additionally, the authors should add a new figure, Figure 3, to illustrate some of the applications of lung-on-a-chip models discussed in Section 3, as this would help conclude the manuscript on a more comprehensive note.
Comments on the Quality of English Language
it can be improved
Author Response
-
Modification of Figure 1 & 2: The Lung-on-a-chip icon was modified in hopes of conveying its complexity in the overview figure. We have revised Figure 2 to specifically represent an overview of a prototypical lung-on-a-chip construction, focusing on its fundamental components and structure. This figure now serves as a detailed visual guide for understanding the core architecture of a typical lung-on-a-chip model.
-
Creation of Figure 3: We have introduced a new Figure 3 to highlight various designs and applications of lung-on-a-chip systems discussed in Section 3. This figure includes:
- 3a: An overview of the electrospun nanofibrous ECM chip, displaying the pneumatic chamber and other critical features of the design.
- 3b: A focus on the impact of pore size within the electrospun ECM, illustrating how it influences barrier protein expression based on results from our research.
- 3c: A visual depiction of the pollutant lung chip systems, highlighting the different cell layers involved in this model.
- 3d: A graphical representation showing the elevation of TNF-α and DAPI in response to varying pollutant concentrations, with these markers serving as indicators of fibrosis.
We believe that the updated Figure 3 provides readers with a comprehensive visual overview, showcasing how these systems are constructed and the significant results they can yield.
Round 3
Reviewer 3 Report
Comments and Suggestions for Authors
The authors addressed my comments
Comments on the Quality of English Languageit can be improved a little bit